# Design of a Broadband Coplanar Waveguide-Fed Antenna Incorporating Organic Solar Cells with 100% Insolation for Ku Band Satellite Communication

**DOI:** 10.3390/ma13010142

**Published:** 2019-12-30

**Authors:** Yadgar I. Abdulkarim, Lianwen Deng, Halgurd N. Awl, Fahmi F. Muhammadsharif, Olcay Altintas, Muharrem Karaaslan, Heng Luo

**Affiliations:** 1School of Physics and Electronics, Central South University, Changsha 410083, China; yadgar.kharkov@gmail.com (Y.I.A.); denglw@csu.edu.cn (L.D.); 2Physics Department, College of Science, University of Sulaimani, Sulaimani 46001, Iraq; 3Department of Communication, Engineering College, Sulaimani Polytechnic University, Sulaimani 46001, Iraq; halgurd.awl@spu.edu.iq; 4Department of Physics, Faculty of Science and Health, Koya University, Koya 44023, Iraq; Fahmi982@gmail.com; 5Department of Electrical and Electronics, Iskenderun Technical University, Hatay 31100, Turkey; olcayaltintas@gmail.com (O.A.); muharrem.karaaslan@iste.edu.tr (M.K.)

**Keywords:** CPW-fed monopole antenna, satellite communication, organic solar cell, P3HT:PCBM

## Abstract

A broadband coplanar waveguide (CPW)-fed monopole antenna based on conventional CPW-fed integration with an organic solar cell (OSC) of 100% insolation is suggested for Ku band satellite communication. The proposed configuration was designed to allow for 100% insolation of the OSC, thereby improving the performance of the antenna. The device structure was fabricated using a Leiterplatten-Kopierfrasen (LPKF) prototyping Printed circuit board (PCB) machine, while a vector network analyzer was utilized to measure the return loss. The simulated results demonstrated that the proposed antenna was able to cover an interesting operating frequency band from 11.7 to 12.22 GHz, which is in compliance with the International Telecommunication Union (ITU). Consequently, a 3 GHz broadband in the Ku band was achieved, along with an enhancement in the realized gain of about 6.30 dB. The simulation and experimental results showed good agreement, whereby the proposed structure could be specifically useful for fixed-satellite-services (FSS) operating over the frequency range from the 11.7 to 12.22 GHz (downlink) band.

## 1. Introduction

Wireless communications have been undergoing rapid development and are highly requested for communication services over long distances that use high data rates [1]. Nowadays, satellite communication is considerably important for utilization in remote areas and for Internet applications, thereby providing stable connectivity and minimum redundancy. In remote areas where it is difficult to establish a grid electricity transmission, the operation of a stand-alone communication service for long-term use is becoming easy with the help of an independent and continuous supply of electricity. Such steady power supply can be realized by means of photovoltaic (PV) technology. PV devices can be successfully utilized to convert sunlight energy into electricity via solar cells comprised of various architectural designs and active materials [2,3,4]. The integration of solar cells with antennas can overcome the problem of autonomous communication systems and helps in reducing design costs. In such applications, data information is collected or transmitted through electromagnetic waves from and to the satellite by means of two-way wireless communication systems. The microstrip antenna is a crucial and key component for these systems, which shows interesting properties such as lightweight, high broadband gain, and low profile [5,6]. In 1995, the first connection of a 2.225 GHz microstrip patch antenna with solar cells was proposed for microsatellites application [7]. The solar cells had to be placed at a reasonable distance from the radiating edges of the microstrip patch in order to avoid performance degradation. The integration of photovoltaic cells with patch antennas could save real estate and reduce the design costs.

A review of the literature showed that various studies, aimed at supplying antennas with solar cells for terrestrial and satellite applications, have been performed [8,9,10]. Along this line, various designs and architectures were proposed to obtain the maximum insolation of the solar cells, such as mesh patch antennas [11,12,13] and transparent antenna conductors [14,15]. It is, however, known that not all these designs are definitely exposed to 100% insolation and the integration of the PV cells are limited to the inorganic solar cells [16,17]. Inorganic solar cells suffer from the high cost and high-energy requirement during the production process. Very recently, organic solar cells (OSCs) have received considerable attention due to their flexibility, ambient solution-processability, low cost, environmentally friendly, and lightweight properties [18,19,20,21,22,23]. 

The compatibility issue is considered to be one of the main challenges that arise during the coupling of solar cells with the antenna so that the antenna does not block the solar cell from functioning properly and so that the performance of the antenna is not compromised with the presence of the solar cell. We anticipate that the use of OSC with CPW-fed monopole antennas can be specifically useful to avoid the possible mismatches that may arise during the incorporation of solar cells with antennas. This is because organic materials are less vulnerable to the attenuations by electromagnetic radiation compared with the inorganic materials. Besides, the active layer thickness of almost all organic solar cells are within the nanometer scale [24,25,26], which is thin enough, compared to the microscale thickness of inorganic solar cells. The most widely used OSC is based on the mixture of organic poly (3-hexylthiophene-2,5-diyl) (P3HT) and methanofullerene (PCBM) as the active layer of the device [27]. These materials possess excellent properties such as chemical stability, high optical absorption, good thermal and electrical property, as well as excellent solution process ability, making them good candidates for the mass production of OSCs [28,29].

In this work, a broadband conventional coplanar waveguide (CPW) printed monopole antenna to be integrated with an organic solar cell is suggested. The designed structure is composed of three layers, where the organic solar cell is placed on top of the antenna in order to obtain 100% insolation. As such, the OSC is fully exposed to sunlight energy, thereby improving the antenna’s performance. The novelty of this work is the proposition of the utilization of a lightweight, nanostructured, and radiation tolerated solar cell that is integrated with patch antennas for satellite applications. These unique features can only be found in the third-generation solar cells, known as organic solar cell (OSC). The designed structure can be specifically useful for fixed-satellite-services (FSS) operating over the frequency range of the 11.7 to 12.22 GHz band. The remainder of this paper is organized as follows: Details on the design structure and OSC incorporated assembly is given in Section 2, while the achieved results are analyzed and discussed in Section 3, followed by the main conclusions, which are drawn in Section 4.

## 2. Methodology

### 2.1. Design Structure and Fabrication

The structure of the proposed CPW antenna is shown in Figure 1. It is comprised of a rectangular patch with a dimension of 15.2 × 6.5 mm^2^ sandwiched between the OSC and substrate, where a gap of 1.5 mm is left between the antenna and the OSC. The solar cells are connected in a series and parallel combinations of a 3 × 4 array in order to yield an overall open circuit voltage of about *V_oc_* = 1.2 V and a short circuit current density of about *I_sc_* = 20 mA·cm^−2^. Since the solar array backside is mounted onto the backside of the antenna, the antenna performance would not interfere with the solar cell transparency. The proposed configuration was purposely chosen to allow 100% insolation of the OSC, thereby improving the performance of the antenna. The patch geometry was purposely chosen in order to be coupled with step load structure in order to provide a wide bandwidth while keeping the antenna gain as high as possible. Thus, we placed the antenna at the backside of OSC, where the antenna face was directed towards the earth. When it comes to space applications, the CPW-fed monopole radiated more in the opposite direction of the OSC, which was towards the earth. Furthermore, the aluminum electrode of the OSC at the back could act as a reflector to redirect the wave propagation density, which helped the antenna provide higher directivity and gain.

In the numerical simulation, the metal layer of the antenna was chosen to be copper metal with a thickness of 0.035 mm and a conductivity of 5.96 × 107 S/m. The utilized substrate was made of IS680 with a thickness of 0.762 mm, dielectric constant of 2.8, and loss tangent of 0.025. The rectangular patch was coupled to the feeding port via a center step-loaded structure with a 50 Ω SMA connector. The outer part of the SMA connector was connected to the ground plane and the inner part was extended through the substrate in order to be connected to the extended strip. The numerical simulation was performed by using the finite integration technique (FIT) based on the high-frequency electromagnetic Computer Simulation Technology (CST) microwave studio. The optimum required dimensions of the proposed structure have been determined by the genetic algorithm approach, in which the antenna was operated in the Ku band frequency range. The other dimensional parameters of the antenna are given in Table 1. The area of the utilized OSC was considered to be the same size as the antenna size. 

### 2.2. Structure of the OSC

The OSC mounted on top of the CPW-fed monopole antenna was composed of three main layers deposited onto a glass substrate as follows: Indium tin oxide (ITO) electrode (20 nm), P3HT: PCBM organic active layer (100 nm), and aluminum electrode (40 nm). The optical and magnetic related parameters of the layers are given in Table 2, while the OSC architecture is shown in Figure 2.

### 2.3. Fabrication and Measurement of the Proposed Antenna 

The antenna structure was fabricated by using a E33 model LPKF prototyping PCB machine at Department of Electrical and Electronics, Iskenderun Technical University, Hatay, Turkey. In our research work, we had a chance to only fabricate the proposed antennas without organic solar cell due to the instrumentation limitation in our labs. A single sided copper covered the IS680 substrate with a thickness of 0.762 mm chosen in the manufacturing process for the proposed antenna. After the fabrication of the antenna, a 50 Ω SMA connector was soldered to the feeding line and ground plane, as illustrated in Figure 3. The experimental investigation was carried out via the Agilent PNA-L N5234A vector network analyzer in an operating frequency ranging from 10 MHz to 43.5 GHz. The VNA was first calibrated by using a proper calibration kit, considering the short circuit, open circuit, and load situations in the frequency range from 5 to 16 GHz. Finally, the antenna was connected to the VNA and the return loss parameter was measured accordingly.

## 3. Results and Discussion

### 3.1. Effect of Organic Solar Cell on the Proposed Antenna Performance

In order to understand the impact of OSC integration on the antenna performance of the proposed design, modeling was performed using the CST microwave studio. Figure 4 shows a comparison between the simulated return loss of the proposed antenna with and without the presence of OSC. It can be seen from the figure that when OSC was included in the proposed CPW-fed monopole antenna, the reflection response (S11) was significantly enhanced. In the case of the OSC absence, the downlink resonance frequency was 14 GHz with minimum return loss of −22 dB, while in the presence of OSC, the return loss was improved to −38 dB and the resonance frequency was shifted by 3 GHz towards the preferable resonance frequency at 11 GHz. This is considered to be an interesting frequency for satellite applications, which is in compliance with that assigned by the International Telecommunication Union (ITU).

Figure 5 shows the simulated and measured results of the antenna gain versus the frequency with and without the incorporation of OSC. One can notice that, at the resonance frequency of 11 GHz, the peak gain without OSC is about 3.76 dB, while that for the OSC incorporated antenna was increased to 5.90 dB. Considering the measured gain without the OSC, it was seen that the antenna gain at the resonance frequency of 11 GHz reached 6.0 dB. The simulation and measured results were found to be in good agreement. It is concluded that, by means of OSC incorporation, the realized gain can be greatly enhanced. In the case of OCS absence, the radiation pattern of the antenna relatively consisted of two symmetrical lobes on both the front and back of the antenna structure. The OSC can act as a reflector to reflect the propagated wave from the back to front of the antenna, which ultimately led to the increase in the antenna directivity (D) and radiation efficiency. Hence, the antenna gain linearly increased with the increase radiation efficiency as follows: G= ηD, where *η* is the radiation efficiency [30].

The simulated and measured antenna radiation patterns for both E-and H-planes with and without OSC are shown in Figure 6. The antenna without OSC has an Omni-directional radiation pattern in both E and H planes. When the OSC was integrated, the antenna produced a directional radiation pattern in both planes (E-plane and H-plane) with more power confined in the main lobes and less power in the back lobes, which ultimately resulted in the increment of the antenna’s directivity. This property makes use of the antenna in being more interested in directional telecommunication applications such as satellite application. All the radiation patterns were monitored at 11 GHz.

### 3.2. Effect of Patch width Variation on S-Parameters

In order to identify the optimum patch width (W5), at which the performance of the antenna is in its best condition, a variation in W5 was performed and the S-Parameter (S11) was measured accordingly. It was observed that the antenna performed the best return loss when the patch width was fixed at 6.5 mm, as shown in Figure 7. The design dimensions shown in Table 1 were kept constant, while the width of the patch antenna was varied from 5 to 6.5 mm in steps of 0.5 mm. It can be seen from Figure 8 that there was no vibration in the position of the resonance peak with the increase of antenna width, while there was a significant change in the shape of the resonance frequency and increased return loss to about −48 dB at the optimum width of 6.5 mm.

### 3.3. Effect of Air Gap Variation on S-Parameters

A parametric study was carried out by changing the air gap distance between the antenna structure and the OSC from 0.5 to 2 mm in steps of 0.5, as shown in Figure 8. One can notice that when the air gap was 1.5 mm, the best antenna impedance matching was achieved, and hence the OSC played an important role in the impedance matching of the antenna.

In order to understand the operating mechanism of the antennas structure, the current distribution of the proposed antenna, with and without a solar cell, is demonstrated in Figure 9.

Table 3 presents the performance of the proposed design in terms of bandwidth, gain, application, and insolation compared to that reported in the literature. It was observed that the proposed design performed well when it came to 100% insolation for satellite communications.

The simulated and measured return loss results for the CPW design are shown in Figure 10. Considering the −10 dB return loss as a reference point, the operating frequency ranges for both results were almost similar. There was been a mismatch between the measured and simulated results, which might have originated from the lab conditions, calibration errors, and fabrication process with the prototyping machine when considering the size of the antenna structure. It can be concluded from the numerical and experimental results that the proposed antenna structure operated in the Ku band frequency regime. Due to the unavailability of OSC fabrication facilities, just the antenna was fabricated and measured without a solar cell. The reason for the difference between the measured and simulated S_11_ results may be due to the fabrication tolerance and/or substrate and soldering losses. The antenna had a relatively lower efficiency below 10 GHz, and with the addition of the solar structure, the antenna efficiency also reduced a bit. Therefore, that caused the realized gain to be reduced to below 10 GHz. As the antenna had a wide bandwidth with a relatively high gain with the simple size, this made the proposed structure suitable (beside satellite application) for many other wireless applications, including environmental monitoring systems and vehicular communication. Therefore, it can be efficiently used for fixed-satellite-services (FSS) in an operating frequency range from 11.7 to 12.22 GHz.

## 4. Conclusions

In this work, a new antenna structure with CPW-fed, incorporated with an organic solar cell, was designed and verified experimentally in the frequency range of 5–16 GHz, which is useful for Ku band satellite communication. This new design provided 100% insolation of the OSC and improved the performance of the antenna. The proposed antenna covered the frequency band ranging from 11.7 to 12.22 GHz downlink, with a 3 GHz bandwidth, which is in compliance with that assigned by the ITU. The simulated results demonstrated that, when the OSC was mounted on top of the proposed CPW-fed monopole antenna, the return loss was significantly enhanced and the realized gain became 6.30 dB. Furthermore, the antenna produced a directional radiation pattern in both planes (E-plane and H-plane). In this study, the effect of patch width variation on S-Parameters was also investigated. The designed antenna was successfully fabricated and it was found that the experimental results were in a good agreement with the simulated ones. The proposed structure was specifically useful in directional telecommunication applications such as satellite applications.

## Figures and Tables

**Figure 1 materials-13-00142-f001:**
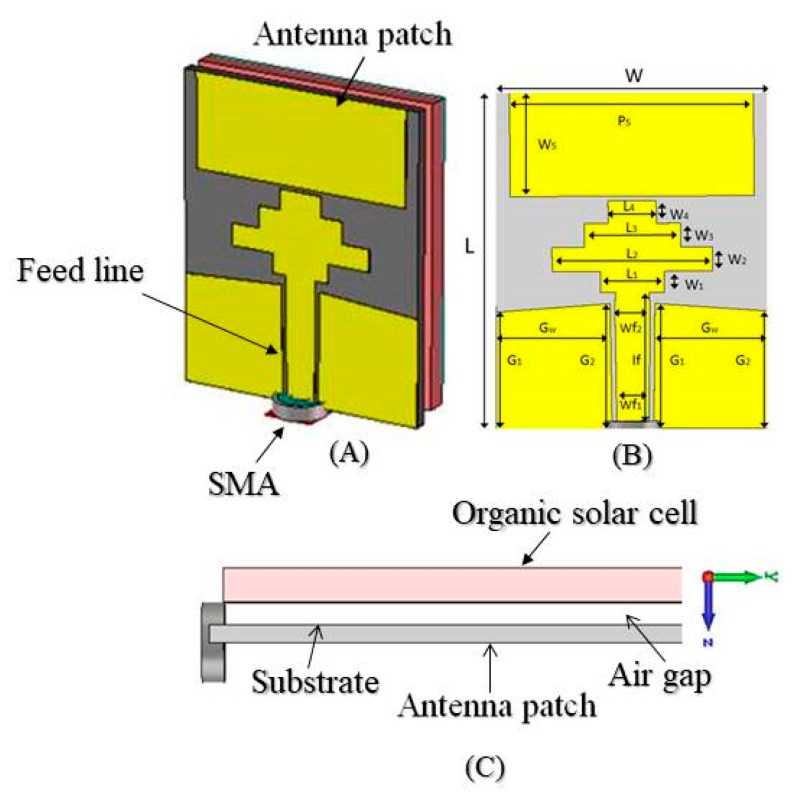
(**A**) The prototype CPW antenna design; (**B**) front view dimensions; and (**C**) side view of the organic solar cell (OSC) integrated antenna.

**Figure 2 materials-13-00142-f002:**
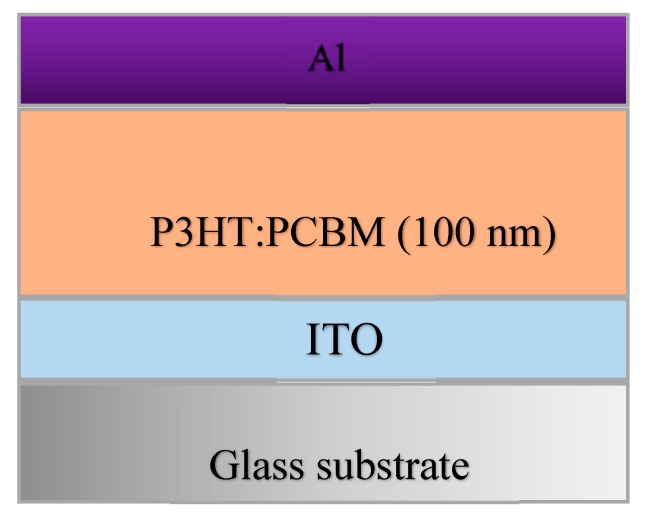
Schematic presentation of the proposed organic solar cell.

**Figure 3 materials-13-00142-f003:**
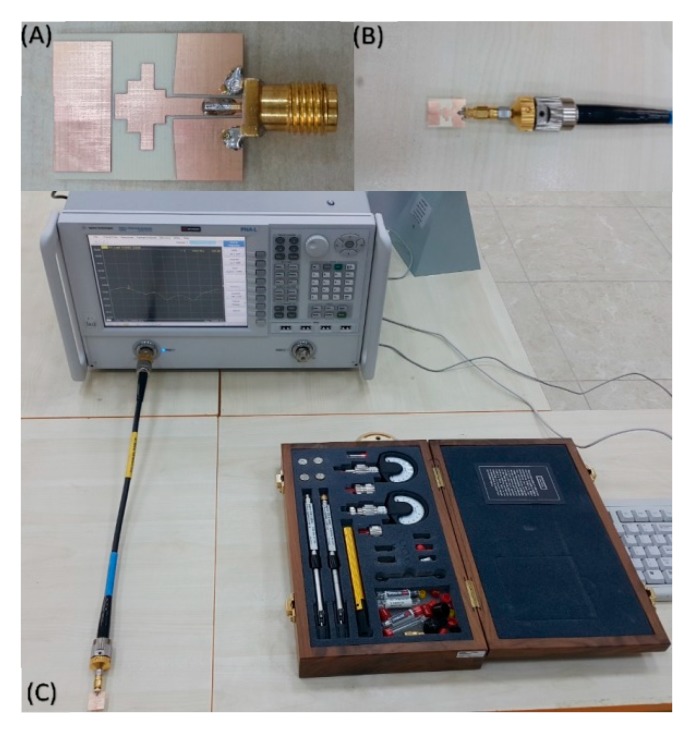
Photograph of the fabricated CPW-fed monopole antenna; (**A**) front view; (**B**) connecting to VNA; and (**C**) experimental set up to measure return loss.

**Figure 4 materials-13-00142-f004:**
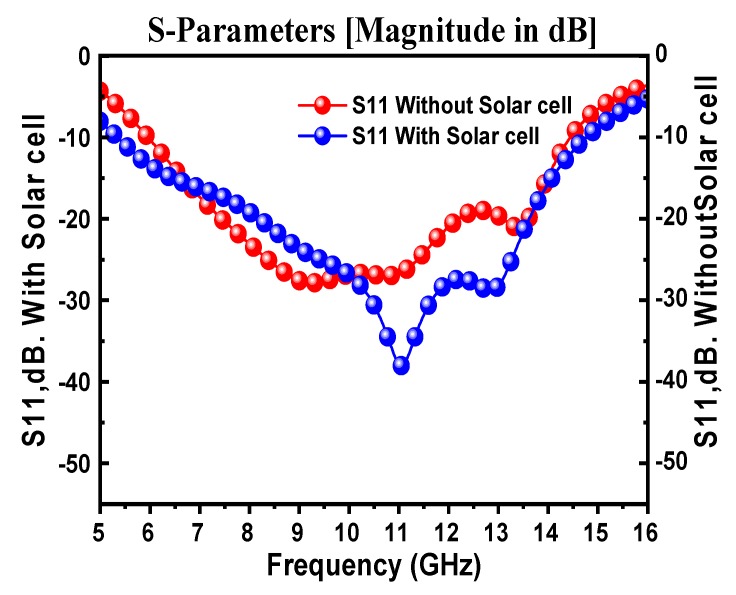
Simulated S11 for the proposed CPW-fed monopole antenna with and without the presence of organic solar cell.

**Figure 5 materials-13-00142-f005:**
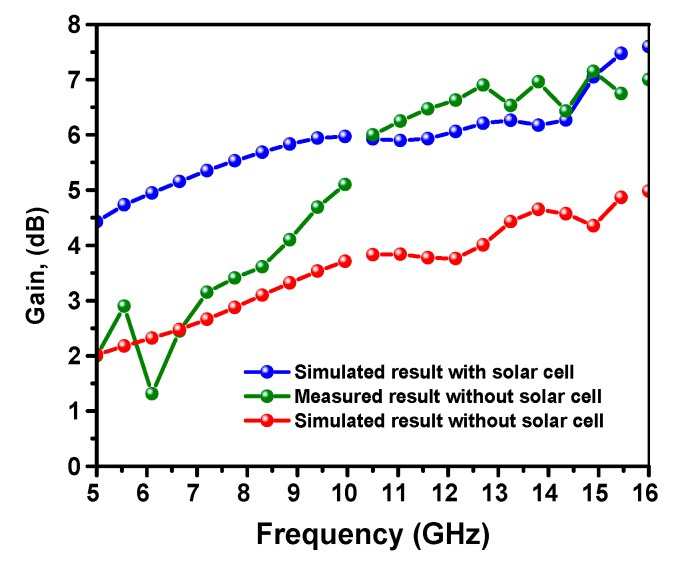
Simulated and measured realized gain of the proposed CPW-fed monopole antenna with and without the presence of organic solar cell.

**Figure 6 materials-13-00142-f006:**
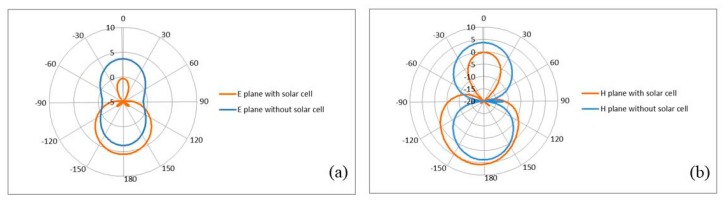
(**a**) Simulated radiation patterns of the proposed antenna with and without OSC downlink for E-plane, (**b**) for H-plane, (**c**) measured radiation patterns of the antenna structure downlink for E-plane and H-plane, and (**d**) simulated 3D radiation pattern for the proposed antenna.

**Figure 7 materials-13-00142-f007:**
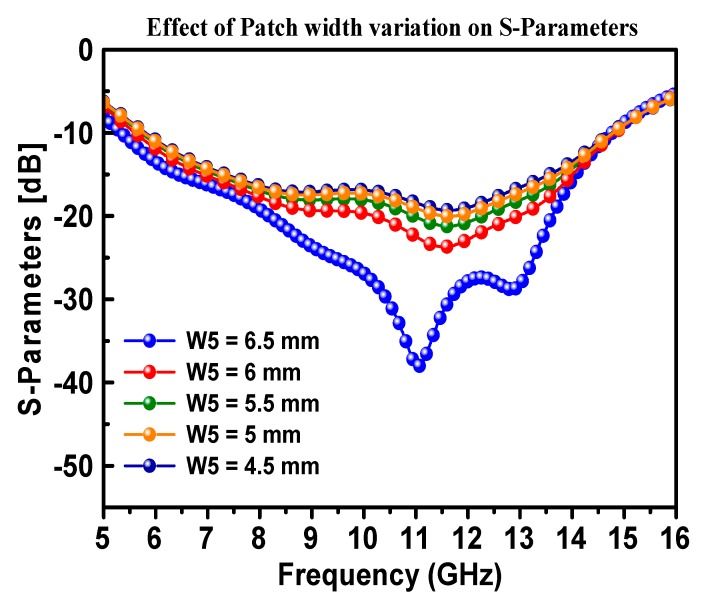
Simulated dependence of the S-Parameters (S11) of the proposed CPW-fed antenna on the changes in patch antenna width (W5).

**Figure 8 materials-13-00142-f008:**
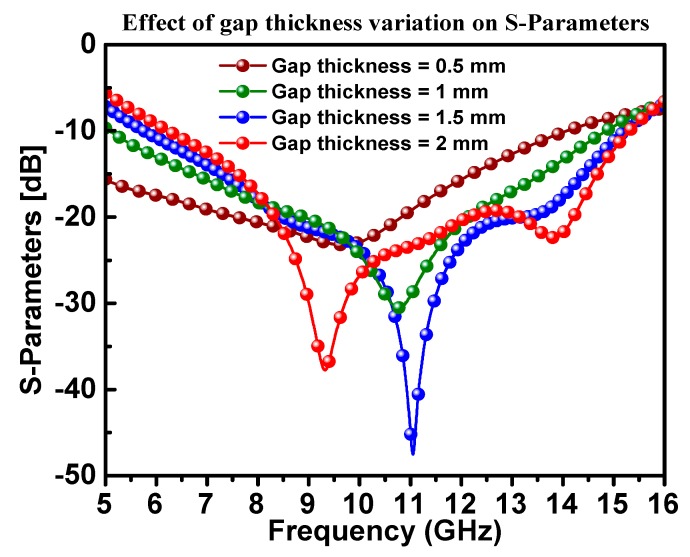
The effect of the antenna on the OSC gap variation on the S11 of the proposed CPW antenna.

**Figure 9 materials-13-00142-f009:**
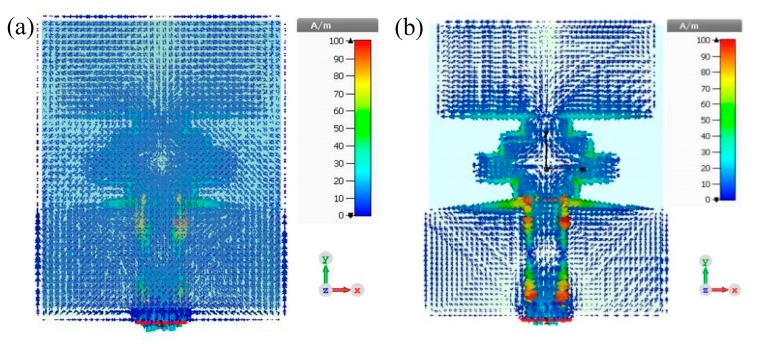
Current distribution of the antenna with OSC (**a**) and without OSC (**b**).

**Figure 10 materials-13-00142-f010:**
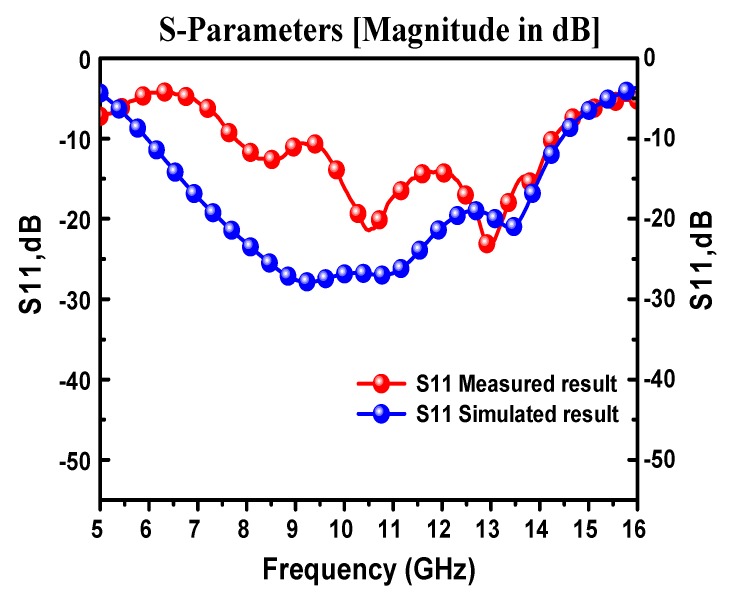
Simulated and measured results of return loss for CPW-fed antenna design.

**Table 1 materials-13-00142-t001:** The dimension related parameters of the proposed CPW antenna used in the simulation design.

Parameters	Value (mm)	Parameters	Value (mm)
W	17	L_2_	1.5
L	21	L_3_	1.5
G_1_	7.8	L_4_	1.4
G_2_	7.3	P_5_	15.2
G_w_	7.27	W_1_	4
Wf_1_	2	W_2_	10
Wf_2_	2.2	W_3_	6
Lf	8.5	W_4_	3
L_1_	1.3	W_5_	6.5

**Table 2 materials-13-00142-t002:** The electrical and magnetic parameters of the OSC integrated with the proposed antenna.

Parameters	P3HT: PCBM	ITO
Relative permittivity	3.920	3.6150
Relative permeability	1.010	1.0001
Bulk conductivity (S/m)	0.300	10^6^

**Table 3 materials-13-00142-t003:** Comparison of the proposed antenna with those reported in the literature.

Reference	Bandwidth (GHz)	Max. Gain (dB)	Application	Insolation %
[9]	2.4–2.5	3.7	Airborne communication nodes, wireless sensor networks.	100%
[10]	2.3–3.2	3	Radio link with solar tracking capability	-
[11]	2.45–2.5	3.5	Wireless sensor network	94.7%
[13]	2.45–2.5	5.15	Small satellite	70%
[15]	10.7–12.7	26.16 dB	Satellite communications	Not 100%
[16]	5.66–5.91	7.8	WiMAX	-
proposed design	5–16	5.90	Satellite communications	100%

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
