# Peer review of "Design of a Broadband Coplanar Waveguide-Fed Antenna Incorporating Organic Solar Cells with 100% Insolation for Ku Band Satellite Communication"

_materials, 2019, doi:10.3390/ma13010142_

Round 1
Reviewer 1 Report
The manuscript shows an antenna with attached a solar cell. The object is interesting, however there are some aspects that should be investigated:
- Since the antenna and the corresponding OSC are very small, they have to be connected in an array. How do you think to connect the solar cells. Does it impact to the performance and the transparency of the structure?
- If I have understood well, the patch antenna radiates more on the opposite direction where the OSC is placed. Since the antenna mounted on a satellite should point through the earth, the solar cell could be not well illuminated by the sun. This reduces the efficiency of the system. Can you comment this?
– It could be interesting to see the currents flowing on the antenna with and without the cell.
-Is there any particular reason for the geometry of the antenna (if any)?
- Have you measured the antenna with the OSC applied?
– The VSWR graph is not really necessary and can be removed as it takes the same information as the |S11| graph.
– Please write the frequency at which the radiation patterns have been computed
Reviewer 2 Report
A CPW-fed monopole antenna with an organic solar cell (OSC) located on top with an air gap is proposed for Ku-band satellite communication. The operating principle and novelty of the proposed antenna is not clear. The authors need to address the followings items for clarity and completeness of the paper.
First of all, the term “microstrip patch antenna” should be replaced by “CPW-fed monopole antenna” because the proposed CPW-fed antenna does not have a ground plane as shown in Figures 1(a) and 1(b). The microstrip patch antenna should have a microstrip structure, but the proposed antenna does not have it. In Figure 1(c), more detailed 3D configuration of the proposed antenna with the exact location and size of the OSC should be provided because the size of the OSC (8.5*11.1 mm^2 ?) is smaller than the monopole antenna and its location affects the performance of the integrated antenna. According to Figure 1(c), it looks like that the OSC is fully cover the monopole antenna, but it says that “The area of the utilized OSC was considered to be 8.5 × 11.1 mm2 mounted on top of the patch antenna.”
In addition, the OSC should be represented as a multilayer structure because it consists of four layers. The OSC consists of four layers. ITO electrode, P3HT:PCBN organic active layer, aluminum electrode, and glass substrate. In fact, the aluminum electrode is good conductor, whereas the ITO electrode is lossy conductor. Therefore, they can block the radiation of the electromagnetic wave. What are the operating principle and the reason for the gain enhancement of the proposed monopole antenna with the OSC on top? It works as a metallic cover or a reflector? The location of the satellite would be above the OSC or below the monopole? It is very confusing.
The analysis of current distribution on the OSC should be provided. In addition, the configuration of the antenna for the E- and H-plane radiation patterns of the proposed antenna in Figure 6 should be provided. The reviewer suspects that the main beam direction (180 degree) is toward bottom of the antenna not top direction. The authors need to double-check this out. In this regard, measured gain and radiation patterns should be included to validate the simulated results. In addition, efficiency also needs to be compared. What is the novelty and contribution of the proposed antenna? It seems like that it is just a simple combination of a well-known CPW-fed monopole antenna and an OSC with an air gap.
The design principle is not clear. How 100% insolation is achieved if the monopole antenna blocks the OSC? What is the advantage of using the OSC instead of a simple metallic reflector? Air gap between the monopole antenna and the OSC is very important parameter to determine the performance of the antenna. Parametric study on this should be investigated. How did you come up with air gap = 1.5 mm? Transmission and reflection coefficient characteristics of the OSC itself should be included.
In Figure 3, the photograph of the complete structure of the proposed antenna with the OSC needs to be added. In Figure 5, gain of the proposed antenna with the OSC is improved only in the frequency range from 10 GHz to 15 GHz, but it is reduced in the low frequency band. What is the reason for this? In Figure 9, the difference between the measured and simulated S11 results is quite large in the low frequency band. What is the reason? The authors need to check this out. In fact, the frequency bandwidth of the proposed antenna is very wide, but the low frequency band is useless for Ku band satellite communication. The reason for using wideband monopole antenna needs to be addressed.
The authors are using S11 and return loss as the same quantity, but they are different. Return loss should be replaced by S11 or input reflection coefficient. Performance comparison with other antennas integrated with solar cells in the literature should be provided. Some of the format in references is not consistent with materials format. The authors need to double-check this out.
Round 2
Reviewer 1 Report
The manuscript is improved compared to the first submission and the authors replied to all the comments.
In the new version there is only a concern about Figure 5. In the text (line 206) the authors say that the measured gain of the antenna at 11GHz is 6.12dBi but looking at the figure it seems lower. Also the double vertical scale used in Figure 5 can be slightly confusing for the reader. This reviewer suggests the use of a single vertical scale for this figure, considering also that all curves represent the gain of the antenna.
Reviewer 2 Report
The revised manuscript reflected some parts of the reviewer’s comment, but several issues did not addressed yet. Therefore, the authors still need to address the followings items for clarity and completeness of the paper.
As mentioned in the previous comment, the term “microstrip” should not be used because the proposed antenna is fed by a coplanar waveguide transmission line not a microstrip feed line. It is more like a CPW-fed monopole antenna with a parasitic patch backed by an OSC reflector. The operating mechanism provided in the response needs to be added in the revised manuscript to help reader’s understanding. Measured S11, gain, and radiation patterns are not provided in the revised manuscript. These should be included to validate the simulated results. In addition, efficiency also needs to be compared. Otherwise, the paper cannot be published. It seems like that the authors do not have the fabricated OSC. The integrated antenna should be fabricated and measurement results should be provided to validated the simulated results. As mentioned in the previous comment, sir gap between the monopole antenna and the OSC is very important parameter to determine the performance of the antenna. Parametric study on this should be investigated. How did you come up with air gap = 1.5 mm? Pls. add the parametric study results for the air gap. The simulated and measured transmission and reflection coefficient characteristics of the OSC itself should be included.
Round 3
Reviewer 2 Report
The authors failed to provide the measured performance of the OSC reflector and the integrated antenna with the OSC reflector. In this regard, the reviewer recommends rejecting the paper and the authors are encouraged to submit the paper after finishing the fabrication of the OSC reflector and the integrated antenna to validated the simulated results.